# PARP Inhibitors: A Major Therapeutic Option in Endocrine-Receptor Positive Breast Cancers

**DOI:** 10.3390/cancers14030599

**Published:** 2022-01-25

**Authors:** Laetitia Collet, Julien Péron, Frédérique Penault-Llorca, Pascal Pujol, Jonathan Lopez, Gilles Freyer, Benoît You

**Affiliations:** 1Oncology Department, CITOHL, Lyon-Sud Hospital, Cancer Institute of Hospices Civils de Lyon (IC-HCL), Hospices Civils de Lyon, 69495 Lyon, France; laetitia.collet@lyon.unicancer.fr (L.C.); julien.peron@chu-lyon.fr (J.P.); gilles.freyer@chu-lyon.fr (G.F.); 2Lyon-Sud Medicine School, University of Lyon, University Claude Bernard Lyon 1, 69008 Lyon, France; 3Laboratoire de Biométrie et Biologie Evolutive, Equipe Biostatistique-Santé, CNRS UMR 5558, Université Claude Bernard Lyon 1, 69100 Villeurbanne, France; 4Department of Pathology and Biopathology, Jean Perrin Comprehensive Cancer Center, UMR INSERM 1240, University Clermont Auvergne, 63011 Clermont-Ferrand, France; frederique.penault-llorca@clermont.unicancer.fr; 5Department of Cancer Genetics, CHU Montpellier, UMR IRD 224-CNRS 5290, Université Montpellier, 34295 Montpellier, France; p-pujol@chu-montpellier.fr; 6Centre de Recherches Écologiques et Évolutives sur le Cancer (CREEC), UMR 224 CNRS-5290, University of Montpellier, 34394 Montpellier, France; 7Biochemistry and Molecular Biology Department, Hopital Lyon Sud, Hospices Civils de Lyon, Université Claude Bernard Lyon 1, 69008 Lyon, France; jonathan.lopez@chu-lyon.fr

**Keywords:** breast neoplasms, homologous recombination, poly (ADP-ribose) polymerase inhibitors

## Abstract

**Simple Summary:**

OlympiAD and EMBRACA trials demonstrated the efficacy of PARPi, compared to chemotherapy, in patients with HER2-negative metastatic breast cancers (mBC) carrying a germline *BRCA* mutation. Patients with ER+/HER2-*BRCA*-mutated mBC seemed to have a higher risk of early disease progression while on CDK4/6 inhibitors and benefit from PARPi, especially when prescribed before chemotherapy. Importantly, the frequency of BRCA pathogenic variant (PV) carriers among ER+/HER2- breast cancer patients has been underestimated, and 50% of all BRCA1/2 mutated breast cancers are actually of ER+/HER2- subtype. Recent studies also highlight the benefit of PARPi in BRCA wild type mBC with HRD representing up to 20% of ER+/HER2- breast cancers. The OLYMPIA trial also demonstrated PARPi utility in patients with ER+/HER2- early breast cancers with *BRCA* PV at high risk of relapse. Consequently, implementation of early genotyping and new strategies for identifying patients with high-risk ER+/HER2- HRD breast cancers likely to benefit from PARPi is of high importance.

**Abstract:**

Recently, OlympiAD and EMBRACA trials demonstrated the favorable efficacy/toxicity ratio of PARPi, compared to chemotherapy, in patients with HER2-negative metastatic breast cancers (mBC) carrying a germline BRCA mutation. PARPi have been largely adopted in triple-negative metastatic breast cancer, but their place has been less clearly defined in endocrine-receptor positive, HER2 negative (ER+/ HER2-) mBC. The present narrative review aims at addressing this question by identifying the patients that are more likely benefit from PARPi. Frequencies of BRCA pathogenic variant (PV) carriers among ER+/HER2- breast cancer patients have been underestimated, and many experts assume than 50% of all BRCA1/2 mutated breast cancers are of ER+/HER2- subtype. Patients with ER+/HER2- BRCA-mutated mBC seemed to have a higher risk of early disease progression while on CDK4/6 inhibitors and PARPi are effective especially when prescribed before exposure to chemotherapy. The OLYMPIA trial also highlighted the utility of PARPi in patients with early breast cancers at high risk of relapse and carrying PV of BRCA. PARPi might also be effective in patients with HRD diseases, representing up to 20% of ER+/HER2- breast cancers. Consequently, the future implementation of early genotyping strategies for identifying the patients with high-risk ER+/HER2- HRD breast cancers likely to benefit from PARPi is of high importance.

## 1. Introduction

Targeting deoxyribonucleic acid (DNA) damage repair (DDR) pathways has recently emerged as a major opportunity for managing cancers of different origins [1]. Indeed, it is considered that most cancer cells exhibit deficiency in one of the five main DDR systems [2], making their survival dependent on the other DDR pathways [3]. The blockage of these latter pathways induces the “synthetic lethality” due to the precipitation of unrepaired DNA damage.

This concept spurred on the development of polyadenosine diphosphate–ribose polymerase (PARP) inhibitors (PARPi) in tumors characterized by deficiencies of the homologous recombination (HR), such as ovarian cancers, breast cancers, prostate cancers, and pancreatic cancers [1]. HR is a DNA repair complex in charge of repairing double-strand DNA lesions. The initially described alterations involved BRCA1 and BRCA2 pathogenic variants (PVs), found in about 20–30% of ovarian cancers at diagnosis, urging the development of PARPi. In other words, the blockage of single-strand DNA lesion repair by PARPi contributes to the accumulation of single and double-strand DNA breaks that cannot be repaired by defective HR, in the case of BRCA1/2 PV or alterations of other genes of the homologous recombination complex. As a consequence, the inhibition of PARP concomitant in patients with diseases associated with HR deficiency results in genomic instability leading to cancer cell death [4]. This strategy was effective in patients with ovarian cancers carrying germline or somatic BRCA1/2 mutations [5,6,7,8,9,10]. Beyond BRCA1/2 mutations, this concept was subsequently extended to alterations of other elements of the pathway, grouped into homologous recombination deficiencies (HRD) [11].

It is now estimated that 10–25% of breast carcinomas are associated with HRD [11,12,13], due to BRCA1/2 PV [14,15,16,17], but also of other genes such as PALB2, ATM, ATR, BARD1, RAD51, BRIP1, or FANC [18,19,20,21]. Because of the higher frequency of BRCA1 PV (~10%) in patients with triple negative breast cancers (TNBC) [22,23], and the limited number of effective treatment options, PARPi have recently been adopted as a valuable asset in this subgroup [24]. However, the place of PARPi in patients with endocrine-receptor positive, HER2 negative (ER+/HER2-) breast cancer has been less clearly defined due to the lower relative frequency of BRCA1/2 PV (~5%) along with more effective lines of treatments available. This makes the benefit related to PARPi less obvious among all other therapeutic options. In particular, the standard first-line treatment in ER+/HER2- mBC patients relies on CDK4/6 inhibitors, which were shown to be associated with overall survival (OS) benefit [24,25,26,27,28]. Several recent data suggest that patients with HRD ER+/HER2- breast cancers, especially those with high-risk features, may derive a large benefit from the integration of PARPi in the disease management algorithms. Interestingly, recent exploratory analyses showed that the PFS of patients with ER+/HER2- mBC carrying BRCA1/2 mutations treated with first-line CDK4/6 inhibitors was lower than those of patients with wild-type BRCA1/2, thereby suggesting that these disease would be less responsive to CDK4/6 inhibitors, and at higher risk of disease progression [29,30].

In the present narrative review, we review the frequency of HR-related gene mutations in breast cancer and the data on PARPi efficacy in patients with ER+/HER2- breast cancers, especially HR+/HER2- breast cancers with high-risk features. We then present the different genomic tests able to identify the patients the more likely to benefit from PARPi, to discuss their place in the management of ER+/HER2-breast cancers.

## 2. Results

### 2.1. Frequency of Homologous Recombination-Related Gene Mutations

#### 2.1.1. Frequency of Germline BRCA Mutations

It is estimated that around 2–5% of patients with ER+/HER2-mBC harbor a germline BRCA1/2 mutation, with a predilection for BRCA2 mutations [31,32]. Indeed, up to 80% of BRCA2 mutations occur in patients with ER+/HER2- breast cancers [31,33].

The prevalence of BRCA1/2 PV was initially estimated with historical data at approximately 1 in 400 in the general population [34,35]. Recent unselected population-genomic screening demonstrated a higher-than-expected prevalence of BRCA1/2 PV in European individuals at about 1 in 200 [36,37]. Moreover, the BRCA2 gene accounted for more than 60% of total BRCA1/2 PV carriers.

In recent studies of BRCA1/2 screening in unselected breast cancer cases (Table 1), the ratio was closer to 50% [38].

Therefore, the frequency of BRCA2 PV carriers among women with breast cancer might have been underestimated, because the family history of breast and ovarian cancer is less marked, and breast or ovarian cancers are less penetrant or delayed in lifetime. Accordingly, the study by Li et al. showed that a higher percentage (81%) of BRCA2 as compared to BRCA1 (46%) PV carriers were missed out by clinical screening depending on family history and triple-negative phenotype mainly [16]. Many experts assume that given the equal proportion of BRCA1 versus BRCA2 mutated breast cancers, and approximately 20% of BRCA1 and 80% of BRCA2 PV are associated with ER+ breast cancers, 50% of all BRCA1/2 mutated breast cancers are of ER+/HER2- subtype.

#### 2.1.2. Frequency of BRCA Somatic Alterations

The frequency of somatic BRCA1 and BRCA2 PV may be between 1 and 3% of patients in unselected breast cancer populations [19,22,39,40,41,42,43,44], including ER+/HER2- breast cancer patients [19,40,42] and even higher in patients with ER+/HER2- mBC [12,21,45]. In addition, silencing methylation of BRCA1 and BRCA2 promoters was reported in 2–4% of patients [39,43,46].

#### 2.1.3. Mutations in Other Homologous Recombination-Related Genes

Germline PV of other genes of the HR, especially PALB2, CHEK2, and ATM, were found in 0.5–1.5% of patients with unselected breast cancers [12,18,19,20,21,39,40,41,42,43,47,48,49] (Table 1), and were obviously more frequent among patients with family breast cancer history [18,50,51,52,53]. In addition, rearrangements of other genes that are not directly involved in DNA repair could also confer PARPi sensitivity, such as CDK12 deficiency [54,55], or the recently described EWS-FLI1 fusion [56].

In total, it is considered that molecular alterations prone to favor the efficacy of PARPi might involve 10–20% of patients with ER+/HER2- breast cancers [57,58].

**Table 1 cancers-14-00599-t001:** Prevalence of somatic and/or germline pathogenic variant in BRCA1/2 and in other homologous recombination-related genes in breast cancers.

	Study Design	*BRCA1*	*BRCA2*	*RAD51*	*RAD50*	*MRE11*	*NBN*	*PALB2*	*ATM*	*ATR*	*BAP1*	*BARD1*	*BRIP1*	*CHEK2*	*FANC*
FAMILIAL BREAST CANCER POPULATION															
Buys et al., Cancer 2017 [59]	35,409 women with breast cancers eligible for genetic counselling	24%	24%	*RAD51C* 1.6% *RAD51D* 0.6%	NR	NR	1.7%	9.3%	9.7%	NR	NR	2%	3.2%	11.7%	NR
	*Other subtypes than triple negative breast cancer* *N = 30,612 (87%)*	17.3%	26.1%	*RAD51C* 1.3% *RAD51D* 0.5%	NR	NR	2%	9.5%	11.6%	NR	NR	1.7%	3%	14.3%	NR
Slavin et al., NPJ Breast Cancer 2017 [18]	2134 *BRCA* negative familial breast cancers	NR	NR	*RAD51D* 0.19% *RAD51C* 0.14%	0.2%	0.05%	0.05%	0.9%	1.5%	NR	0%	0.3%	0.05%	1.6%	*FANCC* 0.05% *FANCM* 0.3%
	*ER+/HER2- breast cancers* *N = 1203 (50%)*	NR	NR	NR	NR	NR	1.8%	0.8%	NR	NR	NR	NR	NR	NR	NR
Tung et al., Cancers 2015 [60]	1781 women with breast cancers eligible for genetic counselling	4.3%	4.8%	NR	NR	NR	NR	0.6%	0.6%	NR	NR	0.03%	0.04%	1.6%	NR
	377 women with breast cancers eligible for genetic counselling and without *BRCA* mutation	NR	NR	NR	NR	NR	NR	0.02%	0.02%	NR	NR	0.02%	0%	1.3%	NR
UNSELECTED PRIMARY BREAST CANCERS															
Kurian et al. J Clin Oncol 2019 [38]	18,601 unselected women with breast cancer	3.2%	3.1%	*RAD51C* 0.18% *RAD51D* 0.12%	NR	NR	NR	1%	0.7%	NR	NR	0.21%	0.22%	1.6%	NR
	*ER+/HER2-* breast cancers *N* = 9740 (52%)	2%	3.2%	*RAD51C* 0.11% *RAD51D* 0.19%	NR	NR	NR	1%	0.9%	NR	NR	0.21%	0.28%	1.7%	NR
Tung et al., J Clin Oncol 2016 [39]	488 primary breast cancers	3.6%	2.4%	*RAD51C* 0.2% *RAD51D* 0.2%	NR	NR	0.2%	0.2%	0.8%	NR	NR	NR	0.8%	2%	NR
	*ER+/HER2- breast cancers* *N = 301 (62%)*	1.7%	3.3%	*RAD51C* 0.3% *RAD51D* 0%	NR	NR	0%	0.3%	1%	NR	NR	NR	0.3%	1.3%	NR
Hu et al., J Natl Cancer Inst 2020 [20]	54,555 early breast cancers	2.2%	2.2%	*RAD51C* 0,2% *RAD51D* 0,1%	NR	NR	0.3%	1%	1.1%	NR	NR	0.3%	0.3%	1.7%	NR
	*ER+/HER2- breast cancers* *N = 26,620 (58%)*	0.9%	2.1%	*RAD51C* 0.2% *RAD51D* 0.1%	NR	NR	0.3%	0.9%	1.1%	NR	NR	0.2%	0.3%	1.9%	NR
Chen et al., Aging 2020 [47]	524 early breast cancers	3.4%	2.1%	*RAD51C* 0.6%	NR	NR	NR	0.7%	0.6%	NR	NR	NR	0.6%	0.4%	*FANCA* 0.4%
	*ER+/HER2- breast cancers* *N = 363 (69%)*	1.1%	4.1%	*RAD51C* 0.3%	NR	NR	NR	0.6%	0.6%	NR	NR	NR	0.6%	0.6%	*FANCA* 0.6%
Wu et al., Cancer 2020 [19]	605 non-triple negative breast cancer samples from TCGA database	Somatic 1.4%	Somatic 1.4%	*RAD51B* 0.5% *RAD51C* 0.3% *RAD51D* 0.3%	0.7%	1%	0.5%	0.7%	2.4%	1.4%	0.5%		1%	0.7%	*FANCA* 0.8% *FANCC* 0.8% *FANCD2* 1.2% *FANCE* 0.3%
Pereira et al., Nat commun 2016 [40]	2433 early breast cancers	1.7%	1.8%	NR	NR	NR	NR	NR	NR	3.8%	1.6%	NR	1%	0.7%	*FANCA* 2.5 *FANCD2* 1.8
	*ER+/HER2- breast cancers* *N = 1563 (64%)*	1%	1%	NR	NR	NR	NR	NR	NR	4%	1%	NR	1%	1%	*FANCA* 2% *FANCD2* 2%
UNSELECTED METEATATIC BREAST CANCER or INCLUDING METASTATIC BREAST CANCERS															
Paul et al., J Clin invest 2020 [49]	66 metastatic breast cancers	4.5%	4.5%	NR	NR	NR	NR	3%	1.5%	NR	NR	NR	NR	1.5%	NR
	*ER+/HER2-; breast cancers* *N = 46 (70%)*	NR	NR	*RAD51C* 0.2% *RAD51D* 0.1%	NR	NR	0.3%	0.9%	1.1%	NR	NR	0.2%	0.3%	1.9%	NR
Rinaldi et al., Plos one 2020 [21]	11,616 breast cancers Somatic pathogenic variant from primaries (39%) lymph nodes (12%) and metastases (43%)	5.6%	7.2%	NR	NR	NR	NR	NR	NR	NR	NR	NR	NR	NR	NR
	*ER+/HER2- breast cancers* *N = 6388 (55%)*	3.4%	8.5%	NR	NR	NR	NR	2.4%	5.4%	5%	NR	NR	NR	2.2%	NR
Angus et al. Nat genet 2019 [12]	442 metastatic breast cancer and metastatic biopsies Results in ER+/HER2- breast cancers *N* = 279 (63%)	2.2%	6.1%	NR	NR	NR	NR	1.1%	6.1%	5.4%	NR	NR	NR	NR	NR

ER+/HER2-: ndocrine receptor-positive HER2 negative; NR: not reported.

### 2.2. PARP Inhibitors in ER+ Breast Cancer

#### 2.2.1. Efficacy in Patients with ER+ BRCA1/2 Mutated Metastatic Breast Cancer

In 2017, OlympiAD was the first trial demonstrating the benefit of olaparib compared to chemotherapy of the investigator’s choice in HER2 negative mBC [61] (Figure 1) (Table 2). Patients had previously received taxane or anthracyclines regimens therapy and at least one prior endocrine therapy for HR+/HER2-mBC patients. Olaparib was associated with a reduced risk of progression of 42% compared to chemotherapy (hazard ratio 0.58, 95% CI 0.43–0.80 *p* < 0.001, absolute benefit of 2.8 months in PFS). In the subgroup analysis, olaparib was not associated with PFS improvement in ER+/HER2- mBC patients (hazard ratio 0.82, 95% CI 0.55–1.26), despite an impressive 65.4% ORR [62]. Of note, a high proportion of them had already received prior chemotherapy (77%). With a 25.3 months median follow-up, no OS benefit was found in the overall population. However, the exploratory subgroup analysis suggested that patients treated with olaparib might have experienced an improved OS when they were naive of chemotherapy (hazard ratio 0.51, 95% CI 0.29–0.90), contrary to those previously treated with chemotherapy (hazard ratio 1.13, 95% CI 0.79–1.64). The OS benefit was consistent whether the tumor was ER+/HER2- (hazard ratio 0.86, 95% CI 0.55–1.36) or TNBC (hazard ratio 0.93, 95% CI 0.62–1.43) [62].

In parallel, the phase III trial EMBRACA study [63] demonstrated a statistically significant improvement of 3 months PFS with talazoparib monotherapy compared to chemotherapy at the investigator choice. Patients had received no more than three previous chemotherapy treatments, and 91% of ER+/HER2-mBC patients had been previously treated by endocrine therapy. The benefit was found in all subgroups, except for patients who had previously received platinum-based treatment. No improvement in OS was observed in the overall population. Interestingly, there was a non-significant trend for OS improvement associated with talazoparib among ER+/HER2- patients who received no more than one prior line of chemotherapy for their mBC (hazard ratio 0.62, 95% CI 0.36–1.04) [64].

The impact of exposure to previous chemotherapy treatment on talazoparib efficacy was corroborated in the phase II ABRAZO trial, since the 29% ORR found in ER+/HER2-mBC patients pretreated with ≥ 3 previous cytotoxic chemotherapy treatments, was much lower than the 63% reported in the EMBRACA trial [64,65].

Of note, both olaparib and talazoparib showed acceptable safety profiles, with adverse events graded 1 or 2 mainly. In addition, patient-reported-outcomes and quality-of-life studies highlighted significant improvements in global health status with PARPi compared to chemotherapy [66,67].

These outcomes were thereafter validated in an observational prospective cohort LUCY, reporting a median PFS of 8.11 months (95% CI 6.93–8.67) in the whole population, and a consistent median PFS of 8.3 months (95% CI 7.60–9.80) in ER+/HER2- patients. The activity of olaparib appeared also higher in patients who were not previously exposed to chemotherapy: median PFS, 8.3 months, 95% CI 7.0–9.7 in patients not previously treated with chemotherapy, and 7.4 months, 95% CI 5.60–8.80 in those previously treated with at least one prior chemotherapy treatment.

The randomized phase III BROCADE3 trial assessed the effects of a chemo-sensitization by a PARPi. Veliparib was combined with a carboplatin and paclitaxel regimen, and then given as maintenance treatment, compared to placebo. Veliparib was associated with a significant absolute PFS gain of 1.9 months (median PFS 14.5 versus 12.6 months, hazard ratio 0.71, 95% CI 0.57–0.88) [68]. Veliparib activity was found in all subgroups, except for patients pretreated with a platinum-based regimen for metastatic disease. Of note, the risk of progression was decreased by 31% in patients with ER+/HER2-mBC (hazard ratio 0.69, 95% CI 0.52–0.92). However, veliparib was not associated with improved OS with a median follow up of 35 months.

Altogether, these results highlighted PARPi efficacy in ER+/HER2- mBC patients carrying germline BRCA1/2 mutations, especially when given earlier in the treatment strategy. Of note, very few patients had previously been treated with CD4K/6 inhibitors and these studies enrolled patients carrying germline BRCA1/2 mutations only, except for the LUCY trial including three patients with somatic BRCA1/2 mutations.

#### 2.2.2. Efficacy in HRD Tumors beyond Germline BRCA1/2 Mutations

The TBCRC 048 phase II trial addressed the utility of targeting HR with olaparib in enlarged populations of mBC patients, with either germline mutations in non-BRCA HR-related genes (cohort 1) or somatic PV of BRCA1/2 (cohort 2). Of note, 90% of included patients had ER+/HER2- mBC, and 97.5% had already been treated with CDK4/6 inhibitors. Among ER+/HER2- mBC patients, the mPFS were 13.3 and 6.3 months in cohorts 1 and 2, respectively, and were not different from the whole population (Figure 1). In patients with germline PALB2 mutated breast cancers, the ORR was 82%, and the median PFS was 13 months. In addition, ORR and median PFS were 50% and 6 months, respectively, in patients with somatic BRCA1/2 mutated tumors [57].

Simultaneously, Gruber and colleagues assessed talazoparib in a small cohort of mBC patients carrying a germline or somatic mutation in HR-related genes (including 12 patients (92%) with ER+/HER2- tumors). Two of three patients, who experienced complete or partial responses (25% ORR), had a germline PALB2 mutation, and one harbored germline CHEK2, FANCA, and PTEN mutations [58].

Later, the RUBY trial assessed rucaparib activity in a small cohort of patients without a germline BRCA mutation and who harbored tumors associated with HRD status, characterized by a loss of heterozygosity (LOH). The ORR was 11%, involving 4 patients out of 37, comprising one patient harboring somatic BRCA2 and germline PALB2 mutations, and three patients with somatic mutations in other HR-related genes [69].

Contrary to these outcomes, a post-hoc analysis of EMBRACA (with talazoparib), did not find any difference in PFS benefit according to the presence of LOH or not, acknowledging that all patients had germline BRCA1/2 mutated tumors [70].

However, due to the low number of patients, these results are hypothesis-generating, and will have to be confirmed in more extensive studies, such as a recently activated study evaluating talazoparib in patients with PALB2 mutated mBC (NCT04756765). European Society for Medical Oncology (ESMO) guidelines were recently updated and recommend considering PARPi monotherapy in patients with germline BRCA or PALB2 mutations after progression while on CDK4/6 inhibitors [71].

LUCY trial and TBCRC048 were both single arm studies and all patients received PARP inhibitors. The BROCADE3 trial assessed chemotherapy in association with veliparib or placebo and the green bar represents PFS of patients receiving veliparib and chemotherapy.

#### 2.2.3. Efficacy in Patients with BRCA Mutated ER+ Early Breast Cancer in Neoadjuvant Setting

The GeparOla phase II trial compared a neoadjuvant olaparib–paclitaxel combination against the paclitaxel–carboplatin association in invasive HER2- eBC characterized by HRD status, defined with Myriad myChoice^®^ HRD test, or with BRCA1/2 mutations. ER+/HER2- eBC were characterized by a high tumor burden (tumor size more than 2 cm, lymph node involvement, or ki67 > 20%). The olaparib and paclitaxel combination arm was associated with a 55.1% pCR rate compared to 48.6% with the standard regimen. In ER+/HER2- diseases, the pCR rate was 52.6% in the experimental arm versus 20% in the control arm. Of note, the olaparib and paclitaxel combination was associated with a higher pCR rate in germline BRCA1/2 mutated ER+/HER2- eBC patients compared to those with BRCA1/2 wild-type HRD ER+/HER2- eBC (12 out of 21 patients (57%) versus 0 out of 8 patients (0%), *p* = 0.018) [72]. The non-randomized phase II trial NEOTALA strengthened the relevance of this strategy. Neoadjuvant talazoparib given alone offered a 53% pCR in BRCA1/2 mutated eBC. Among five ER+/HER2- eBC patients, three experienced a pCR (60%), including one with a lobular breast cancer [73]. These results are even better than the pCR rate of 46% obtained in patients with early BRCA1/2 mutated TNBC in NEOTALA study, consistent with recent study also assessing neoadjuvant talazoparib in monotherapy [74].

#### 2.2.4. Efficacy in Patients with BRCA Mutated ER+ Early Breast Cancer in Adjuvant Setting

Recently, the large phase III Olympia trial investigated the benefit associated with olaparib given for a year as an adjuvant treatment in HER2- eBC patients with germline BRCA1/2 PV. Patients with ER+/HER2- eBC had unfavorable features (more than four positive lymph nodes for those with initial surgery, or non-pCR and a CPS+EG score ≥ 3 in those treated with neoadjuvant therapy). In the subgroup analysis, olaparib was associated with an absolute increase of 19% of 3-years invasive DFS in ER+/HER2- breast cancer patients who received neoadjuvant chemotherapy (86% vs. 67%, hazard ratio 0.52, 95% CI 0.25–1.04), although not statistically significant probably because of the low number of patients (*N* = 196, 11% of all patients) [75].

These data suggest that PARPi could be of great interest earlier when breast cancer is still in a curative setting in patients with germline BRCA1/2 PV. In June 2021, the American society of clinical oncology (ASCO) updated their guidelines to recommend 1 year of adjuvant olaparib in ER+/HER2- eBC with at least four involved axillary lymph nodes or with residual disease and a CPS+EG score ≥ 3 in the case of previous neoadjuvant chemotherapy [76].

### 2.3. Patient Selection for PARP Inhibitors

#### 2.3.1. Identification of Gene Alterations

Olaparib and talazoparib are approved for patients with HER2- mBC carrying BRCA1/2 germline mutations. As a consequence, the National Comprehensive Cancer Network (NCCN) and the ESMO guidelines recommend the assessment for germline BRCA1/2 mutations in patients with mBC as soon as possible at diagnosis [24,77]. However, no specific companion test has officially been validated in this setting. The National Institute for Health and Care Excellence (NICE) and NCCN acknowledged the need for highly sensitive assays to identify large genomic rearrangements [78,79].

The BRCAnalyse Myriads genetic test, composed of a quantitative PCR, CGH-microarray and bi-directional sequencing, was used in OlympiAD and EMBRACA trials with high sensitivity and specificity for determining BRCA1/2 PV, although it misses some defects such as RNA transcript processing or balanced rearrangements [80,81,82] (Table 3).

In addition to BRCA1/2 status, ESMO ABC5 2020 recommendations advocating for considering other HR-related genes, such as PALB2 [71], is the first official effort to enlarge the PARPi target population. However, gene panel strategies are limited by many uncertainties about the actual impact of many unknown HR-related gene alterations not included in these panels and does not solve the issue of variants of unknown significance (VUS) [83].

#### 2.3.2. Genomic Scars and Genomic Instability

In that context, assessing the DNA genomic scars induced by defective DDR systems, instead of gene alterations, is an attractive approach. Several studies demonstrated that quantification of large-scale state transitions (LST) [84], LOH [85], and telomeric allelic imbalances (TAI) [86] were associated with higher probability of BRCA1/2 mutation, especially BRCA1 mutations [87]. Mutational signatures have also been associated with HRD and BRCA1/2 mutations [12,39].

Available commercial and industrial genetic tests to detect genomic instability

Currently, two commercial tests have been developed including FoundationOne CDx test for BRCA PV and MyriadMychoice for BRCA PV and genomic instability.

FoundationOne CDx test (Foundation Medicine, Cambridge, MA, USA) combines BRCA1/2 PV and percentage of genomic LOH to provide a score (considered as high when ≥ 16), alongside assessment of other HR-related gene alterations.

MyriadMyChoice genomic test associates BRCA1/2 PV, and the three biomarkers LST, LOH, and TAI combined together in a genomic instability score (GIS), categorized as low when <42, or high when ≥42. LST, LOH, and TAI genomics scars detection using MyriadMyChoice genomic test were found in TNBC with BRCA1/2 mutations, and then in the other breast cancer subtypes including ER+/HER2-. Of note, the mean HRD score was found to be similar in both ER+/HER2- and TNBC with BRCA1/2 mutations (mean around 14.5). In addition, the combination of these three individual biomarkers exhibited better predictive value regarding BRCA1/2 deficiency than each of them considered alone [88]. This assay is now recognized by the Food and Drug Administration for assessing the HRD status and the utility of PARPi in patients with advanced ovarian cancers [9,10].

Academic genomic tests to detect genomic instability and mutational signatures

Academic tests use additional information from HR mutational signatures in combination with genomic scars and genomic instability.

HRDetect [89] computes five weighted parameters, including microhomology-mediated small insertions and deletions (indels), GIS, single-base substitution (SBS) signature 3, rearrangement signature 3 and 5 [39]. The sensitivity of HRDetect for predicting BRCA1/2 mutation in the validation cohort was 86% in ER+/HER2- breast cancer patients [89] and showed sensitivity to detect PALB2 biallelic inactivation or RAD51C hypermethylation, and to reclassify BRCA1/2 VUS as germline PV [90]. A prospective study showed that HRDetect was predictive of rucaparib efficacy in a neoadjuvant setting [91].

The random forest-based Classifier of Homologous Recombination Deficiency (CHORD) was developed with the data from patients with metastatic solid tumors, including 19% of mBC. The score integrating a single nucleotide variant, indels, and structural variants, identified HRD status in 24% of primary and 12% of metastatic lesions from breast cancers. Of note, it distinguishes “BRCA1-type HRD”, associated with BRCA1 PV along with deficiency in BRCA1 binding proteins such as BRIP1, FAM175A, FANCA, and BARD1, and “BRCA2 type HRD” associated with PALB2, RAD51C, and BRCA2 PV [13].

In parallel, Bertucci and colleagues combined SBS signature 3 and LST and found a larger proportion of HRD tumors among patients with ER+/HER2- breast cancers. HRD status was found to be more frequent in ER+/HER2- mBC compared to ER+/HER2- eBC (15% versus 8.0%, respectively, *p* = 0.005) [42].

Whilst genomic instability and HRD scores can be seen as potential predictive biomarkers of PARPi efficacy, they do not provide direct information about the origin of HRD. Furthermore, HRD score was initially developed for maximizing the likelihood of BRCA1 mutations [87] and additional work is needed to optimize this tool to ER+/HER2- mBC known to be associated with a higher rate of BRCA2 or PALB2 PV and to harbor specific genomic features such as larger deletion and microhomology [92].

Finally, the main drawback of genomic scar signatures is the lack of consideration of the HRD dynamics or reversion, explaining the reduced value in multi-treated patients [93] whose tumor may change the HR function, which may not be captured by genomic scars, that are indelible [94].

Detection of genomic instability through copy number alterations

Comparative genomic hybridization arrays were used to characterized copy-number (CN) profile of BRCA1/2 mutated breast cancers [95,96] and predict the benefit from chemotherapy [97]. Thereafter, specifically designed MLPA determined the CN profile of up to 50 different genomic regions [98] and demonstrated a good sensitivity and specificity to detect BRCA1-like tumors and predict the response to chemotherapy [97,99]. Then, digitalMLPA allowed to identify the CN profile of up to 700 genomic locations and distinguish non-BRCA-like, BRCA1-like, and BRCA2-like breast cancers. The accuracy was 91% and 82% for the BRCA1-like and BRCA2-like classification, respectively. Moreover, this test may also identify patients with triple-negative or ER+/HER2- breast cancers who could benefit from adjuvant chemotherapy [100]. An on-going phase III trial is assessing a combination of cytotoxic chemotherapy with olaparib in eBC patients with BRCA1-like tumors identified with this digital MLPA assay (NCT02810743).

#### 2.3.3. Functional Homologous Recombination Deficiency and RAD51 Foci Assay

RAD51 recruitment DNA breaks are a hallmark of HR pathways, that immunofluorescence can detect on formalin-fixed paraffin-embedded tumor samples.

RAD51 foci deficiency was significantly associated with a higher Myriad HRD score or biallelic inactivation of HR-related genes including BRCA1, BRCA2, CHEK2 [101], and PALB2 [102]. Functional HRD deficiency was correlated with PARPi and platinum-based chemotherapy efficacy and the subsequent resistance to these drugs in patients carrying the BRCA1/2 mutation [93,102,103,104]. These data suggested that RAD51 foci detection is a dynamic test that can diagnose HRD, and then restored pathways. However, this test cannot detect alterations occurring downstream from RAD51 intervention, such as ATM alterations.

**Table 3 cancers-14-00599-t003:** Different tools to identify the patients who could benefit from PARP inhibitors.

Biomarkers	Resources	Clinical Assessment	Advantage	Limitation
BRCA1/2 pathogenic variant	Targeted sequencing for single nucleotide variant and small indels PCR multiplex for large deletion and duplication	BRACanalyse Myriad Genetic test Phase III clinical trials: OlympiAD [62], Embraca [64], and Brocade3 [68] in metastatic HER2- breast cancer	Easy to perform Validated in clinical trials	BRCA testing only No detection of functional silencing methylation of BRCA gene promoters and of balanced rearrangement (i.e., inversion) No information about variant of unknown significance Patented commercial test cost outsourced
Pathogenic variant of genes of homologous recombination beyond BRCA	Targeted sequencing	Phase II clinical trial for germline PALB2, CHEK2, and FANCA mutation and somatic BRCA1/2, ATR, and PTEN mutations [57,58] in metastatic breast cancer	Easy to perform Validated in clinical trials	Dependence on the genes assessed in the panel, and on knowledge of their implication No detection of functional silencing methylation of gene promoters (i.e., RAD51C) No information about variant of unknown significance Cost
Mutational signatures	Whole exome sequencing	Single base substitution signature 3 Rearrangement signature 3 and Rearrangement signature 5 Preclinical studies [92]	Identification of genomic scars independently of what genes are mutated Identification of genes potentially implicated in HRD and reclassification of variant of unknown significance	Low specificity: different mutational signature and rearrangement signature in function of the homologous recombination related mutated gene Overlook HRD as a dynamic process, persistence of genomic signature despite restoration of HRD missing potential PARP inhibitor resistance Whole exome sequencing could be difficult to perform in daily clinical practice
HRD score (TAI, LOH, LST)	Whole exome sequencing	MyriadMychoice genetic test Phase II clinical trials [69,105]	Validated in clinical trials Identification of genomics scars independently on involved genes Identification of genes potentially implicated in HRD and reclassification of variant of unknown significance	No integration of time, or impact of previous exposure with chemotherapy lines on homologous recombination activity Patented commercial test Cost Limited access to the assay/outsourced
HRDetect (micro-homology mediated indels, HRD index, single base substitution signature 3, rearrangement signature 3 and 5)	Whole genome sequencing	Ad hoc analysis from phase II clinical trial triple negative breast cancer [91]	Identification of genomics scars independently on involved genes Identification of genes potentially implicated in HRD and reclassification of variant of unknown significance	No integration of time or impact of previous exposure with chemotherapy lines on homologous recombination activity No validation in prospective clinical trial Cost Limited access to the assay (research)
Classifier of Homologous Recombination Deficiency (CHORD) (single nucleotide variant, indels and structural variant)	Whole genome sequencing	In vitro studies only	Identification of genomics scars independently on involved genes Identification of genes potentially implicated in HRD and reclassification of variant of unknown significance Differentiation of “BRCA1-type HDR” and “BRCA2-type HRD”	No integration of time, or impact of previous exposure with chemotherapy lines on homologous recombination activity No validation in prospective clinical trial Cost Limited access to the assay
RAD51 foci immunohistochemistry	Fluorescent or chromogenic immunohistochemistry on FFPE samples	Retrospective study and preclinical study Ad hoc analysis from phase II clinical trial triple negative breast cancer [93]	Reduced cost and high feasibility during pathology assessment Real time assessment of homologous recombination activity	No validation in prospective clinical trial Limited to the homologous recombination pathways above RAD51

HRD—homologous recombination deficiency; PARP—polyadenosine diphosphate–ribose polymerase; TAI—telomeric allelic imbalances; LOH—loss of heterozygosity; LST—large scale state transitions; FFPE—Formalin Fixed Paraffin Embedded.

## 3. Discussion

In the past few years, the development of CDK4/6 inhibitors has been associated with a dramatic survival progress in ER+/HER2- mBC becoming the standard treatment in the first-line setting in endocrine sensitive or resistant patients [24]. Nevertheless, between 15 and 30% of patients do not respond to CDK4/6 inhibitors and experience disease progression within 24 weeks of treatment. A growing bulk of data suggests that patients with BRCA1/2-mutated ER+/HER2- mBC are at higher risk of early disease progression while on CDK4/6 inhibitor. A recent retrospective study demonstrated a significantly shorter OS for patient harboring germline BRCA mutations compared to those with wild type BRCA disease treated with CDK4/6 inhibitor and endocrine therapy combination (stratified HR 1.5 95% CI 1.06–2.14). Of note, time to subsequent therapy or death was also shorter for these patients although not significant (stratified HR 1.24, 95% CI 0.96–1.59) [29]. In line with these results, a subgroup analysis of the PADA-1 trial showed that patients with BRCA mutated disease treated with palbociclib and aromatase inhibitor experienced shorter median PFS compared to those with wild type diseases: 14.3 versus 26.7 months [30]. Therefore, PARPi represents a major treatment option in these patients with a higher benefit/risk ratio than chemotherapy, as recently recognized by ESMO [71]. Consistently, the positive outcomes of the OLYMPIA trial suggest that PARPi will become a standard adjuvant treatment in patients treated with bulky or aggressive localized ER+/HER2- cancers, at high risk of relapse leading to a recent update of ASCO guidelines [76].

However, the broader implementation of PARPi used in ER+/HER2- cancers will require addressing some issues. First, the timing of germline BRCA1/2 genotyping will have to be adjusted. In patients with localized ER+/HER2- cancer associated with bulky node involvement after initial surgery, or treated with neoadjuvant chemotherapy, BRCA1/2 status genotyping should be implemented, so olaparib can be prescribed as an adjuvant treatment in patients carrying BRCA1/2 PV. In patients with metastatic diseases, germline BRCA1/2 genotyping should probably be initiated early when the endocrine therapy combined with CDK4/6 inhibitor is started. All available data suggest that PARPi seem to be more effective in patients not exposed to previous chemotherapy. Going further, a phase I study is recruiting to assess Olaparib in combination with Palbociclib and Fulvestrant in ER+/HER2- BRCA1/2 mutated mBC, including, among others, patients in a first-line setting (NCT03685331) and suggesting an even earlier BRCA1/2 genotyping testing.

One may argue that the low frequency of germline BRCA1/2 PV (<5%) in these patients limits the relevance of this strategy. However, the exact rate of BRCA1/2 PV may be higher in patients with ER+/HER2- breast cancer, particularly because BRCA2 PV carriers frequently do not fully fill in personal or family testing criteria in approximately 50% of cases [106]. This proportion was 81.0% in the study by Li et al., as compared to 46% for patients with BRCA1 PV [16]. Furthermore, the potential target patient population for PARPi prescription might be significantly enlarged by increasing the number of assessed molecular alterations. The data from several studies demonstrated that integrating somatic PV of BRCA1/2, silencing methylations of BRCA1 promoters, along with alteration of other genes of the HR pathway such as PALB2, ATM, or CHEK2 may help to identify additional patient candidates for PARPi, representing up to 20% of patients with ER+/HER2- breast cancer. Another option relies on tests assessing DNA scars, such as HRDetect or CHORD, able to integrate the effects of multiple known or unknown gene alterations of the DDR system in one single assay. However, the users of these tests must be aware that DNA scars are definitive and do not detect HR reversion. To overcome this limitation, the use of RAD51 foci as a dynamical companion test appears promising, as it would enable assessment of real-time HR activity, at low cost.

Other issues have no answers yet. The actual OS benefit in patients treated with PARPi is unclear. Similarly, outside TBCRC-048 there are no strong data about the efficacy of PARPi in patients previously treated with CDK4/6 inhibitors, which is the standard first-line treatment of patients with ER+/HER2- mBC. While the actual benefit on OS from PARPi is still uncertain, the benefit/toxicity ratio of this class of drug appears favorable compared to chemotherapy, thereby confirming PARPi as a major option in patients with high-risk HRD ER+/HER2- breast cancers.

## 4. Conclusions

In the past few years, PARPi have emerged as a major therapeutic opportunity in multiple tumor types including breast cancers. Several phase III clinical trials demonstrated the efficacy of PARPi in ER+/HER2- in early and mBC associated with germline BRCA mutation, especially in those with high-risk features characterized by bulky diseases and residual cancer cells after neo-adjuvant chemotherapy in localized BC, and early disease progression during treatment with CDK4/6 inhibitor in mBC. Beyond BRCA mutations, extensive data suggest that PARPi could be effective in a broader population of patients harboring HRD, representing up to 20% of ER+/HER2- breast cancer patients. However, some issues are still unsolved: what companion diagnostic tests should be performed early enough to identify these patients? What is the actual efficacy of PARPi after CDK4/6 inhibitor therapy? Additional studies are warranted to properly answer these questions.

## Figures and Tables

**Figure 1 cancers-14-00599-f001:**
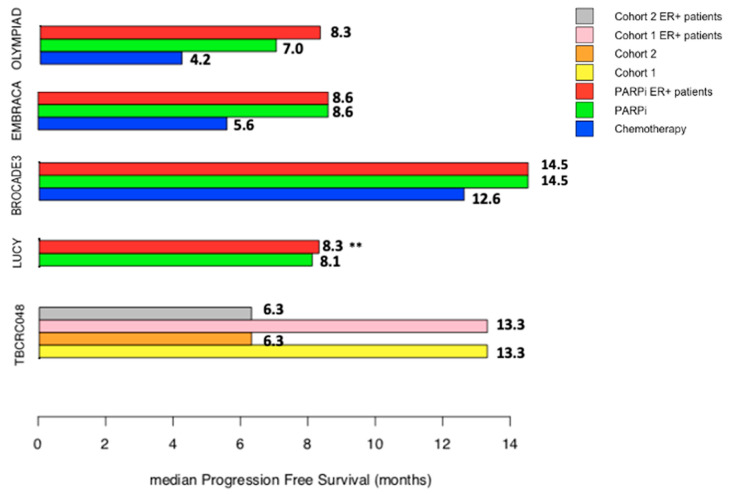
Median progression-free survivals in prospective studies of PARP inhibitors in patients with HER2 negative and HRD-associated metastatic breast cancers.Bar plot showing progression free survival (PFS) in months in different subgroups. In grey, PFS in the endocrine-receptor positive HER2 negative (ER+/HER2-) breast cancer patient subgroup from cohort 2 of the TBCRC048 study corresponding to patients with somatic pathogenic variant of *BRCA1/2* genes or in other DNA repair genes. In pink, PFS in the ER+/HER2- breast cancer patient subgroup from cohort 1 of the TBCRC048 study corresponding to patients carrying a germline mutation in a DNA repair gene other than BRCA. In orange, whole cohort 2 of the TBCRC048 study. In yellow, whole cohort 1 of the TBCRC048 study. In red, PFS in the ER+/HER2- breast cancer patient subgroup with PARP inhibitor treatment. In green, PFS in the whole cohort with PARP inhibitor treatment. In blue, PFS in the whole cohort with chemotherapy alone. ** LUCY trial: Single arm study assessing Olaparib in monotherapy in real life, no placebo group.

**Table 2 cancers-14-00599-t002:** Outcomes of the main clinical trials of PARP inhibitors in HER2- and HRD-associated metastatic breast cancer patients.

	OLYMPIAD	EMBRACA	BROCADE3	LUCY	TBCRC 048
Study design	Phase III randomized *N* = 302	Phase III randomized *N* = 431	Phase III randomized *N* = 509	Phase IIIb single arm *N* = 256	Phase II single arm *N* = 54
Overall population					
Population	Germline BRCA1/2 mutations ≤2 previous cytotoxic regimens for advanced breast cancer Previous taxane and/or anthracycline DFI > 12 months after platinum treatment No limit of previous endocrine therapy, unless one prior endocrine therapy	Germline BRCA1/2 mutations ≤3 previous cytotoxic regimens for advanced breast cancer Previous taxane and/or anthracycline DFI > 6 months after platinum treatment No limit of previous endocrine therapy	Germline BRCA1/2 mutations ≤2 previous cytotoxic regimens for advanced breast cancer Previous taxane allowed but given more than 6 or 12 months before study start in (neo)adjuvant or metastatic setting, respectively DFI > 12 months after platinum treatment No limit of previous endocrine therapy	Germline or somatic BRCA1/2 mutations ≤2 previous cytotoxic regimens for advanced breast cancer Previous taxane and/or anthracycline DFI > 12 months after platinum treatment No limit of previous endocrine therapy, unless one prior endocrine therapy	Germline or somatic mutations in DNA repair gene other than BRCA1/2 (cohort 1) Or somatic pathogenic variant of BRCA1 or BRCA2 genes or in other DNA repair genes ^(1)^ (cohort 2) No limit of prior cytotoxic regimen or endocrine therapy for advanced breast cancer DFI > 12 months after platinum treatment
BRCA testing	Central testing with BRCAnalysis Myriads genetics	Central testing with BRCAnalysis Myriads genetics	Central testing with BRCAnalysis Myriads genetics	BRCA mutation testing in certified laboratory	Genomic profiling of metastatic tumor tissue or blood
PARP inhibitors, experimental arm	Olaparib 300 mg twice daily continuously	Talazoparib 1 mg once daily continuously	Carboplatin + paclitaxel + veliparib 120 mg twice daily on days 2–5	Olaparib 300 mg twice daily continuously	Olaparib 300 mg twice daily continuously
Control arm treatment	Chemotherapy of choice of investigator among capecitabine, eribulin, or vinorelbine No crossover allowed	Chemotherapy of choice of investigator among capecitabine, eribulin, gemcitabine, or vinorelbine No crossover allowed	Carboplatin + paclitaxel + placebo Crossover allowed	NA	NA
Prior chemotherapy n (%)	215 (71%)	265 (61%)	96 (18.8%)	115 (45%)	44 (81%)
Prior platinum n (%)	86 (28%)	76 (17%)	43 (8%)	81 (32%)	3 (6%)
ORR (%)	59.9% versus 28.8% in control arm	62.6% versus 27.2% in control arm	75.8% versus 74.1% in control arm	48.6%	Cohort 1 29.6% Cohort 2 38.5% gPALB2 mutation: 82% sBRCA1/2 mutations: 50%
Median time to response	1.5 months	2.6 months	NR	NR	NR
PFS	7.0 vs. 4.2 months Hazard ratio 0.58 (95% CI 0.43 to 0.80) *p* < 0.001	8.6 vs. 5.6 months Hazard ratio 0.54 (95% CI 0.41–0.71) *p* < 0.001	14.5 versus 12.6 months Hazard ratio 0·71 (95% CI 0.57–0.88) *p* = 0·0016	8.11 months (95% CI 6.93–8.67) No comparison, single arm	Cohort 1: 13.3 months (90%CI 12—NA) Cohort 2: 6.3 months (90%CI 4.4 months—NA) No comparison
OS	19.3 vs. 17.1 months Hazard ratio 0.90 (95% CI 0.66–1.23) *p* = 0.513	19.3 versus 19.5 months Hazard ratio 0.85 (95% CI 0.670–1.073) *p*= 0.17	33.5 vs. 28.2 months Hazard ratio 0.95 (95% CI 0.73–1.23) *p* = 0·67	NR	NR
PARP inhibitor after progression in control arm	8.2%	25%	44%	NA	NA
ER+/HER2- patients					
Number (%)	152 (50.3%)	241 (56%)	266 (53%)	131 (51%)	41 (76%)
ORR (%)	65.4% vs. 36.4%	63.2% vs. 37.9%	NR	NR	30%
Median PFS with PARP inhibitors, Hazard ratio compared to control arm ^(2)^	8.3 months Hazard ratio 0.82 (95% CI 0.55–1.26)	8.6 months Hazard ratio 0.47 (95%CI 0.32–0.71)	14.5 months Hazard ratio 0.69 (95%CI 0.52–0.92)	8.3 months NA	Median PFS 13.3 months for gPALB2 mutation and 6.3 months for sBRCA1/2 mutations NA
Median OS with PARP inhibitors Hazard ratio compared with control arm	21.8 versus 21.3 months Hazard ratio 0.86 (95% CI 0.55–1.36)	NR Hazard ratio of 0.827 (0.56–1.14)	Median OS of 32.4 vs. 27.1 months NR	NR	NR
Previous endocrine therapy n (%)	136 (45%)	219 (91%)	91 (34%)	NR	NR
Prior chemotherapy n (%)	117 (77%)	NR	63 (23.6%)	NR	NR
Prior platinum n (%)	35 (23%)	NR	NR	NR	0 (0%)
Prior CDK4/6 inhibitors n (%)	NR	22 (9%)	NR	NR	40 (97.5%)

DFI—disease free interval; NA—not applicable; PARP—polyadenosine diphosphate–ribose polymerase; ORR—overall response rate; gPALB2—germline PALB2; sBRCA1/2—somatic BRCA1/2; NR—not reported; PFS—progression free survival; OS—overall survival; ER+HER2-—endocrine-receptor positive HER2 negative. ^(1)^ DNA repair genes: ATM, ATR, BAP1, BARD1, BLM, BRIP1, CHEK1, CHEK2, CDK12, FANCA, FANCC, FANCD2, FANCF, MRE11A, NBN, PALB2, RAD50, RAD51C, RAD51D, or WRN. ^(2)^ Median PFS in control arm not reported in the study.

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
