# Peer review of "PARP Inhibitors: A Major Therapeutic Option in Endocrine-Receptor Positive Breast Cancers"

_cancers, 2022, doi:10.3390/cancers14030599_

Round 1
Reviewer 1 Report
First of all,I consider very interesting PARP inhibitors as therapeutic option in endocrine-receptor positive breast cancers, since it has always been considered for the triple negative profile.
A very interesting fact is that the PFS of patients with ER+/HER2- mBC carrying BRCA1/2 mutations treated with first-line CDK4/6 inhibitors was lower than those of patients with wild-type BRCA1/2. Therefore, this benefit should be more specified to know its true magnitude and also quantify this benefit.
It should also be specified that the efficacy of these PARPi is limited only to a benefit in the risk of progression, not finding a benefit in overall survival. Furthermore, it is the patients who have not previously received chemotherapy treatment who show this benefit. Finally, we still do not know the benefit after progression to CDK inhibitors. Although the European Society for Medical Oncology (ESMO) guidelines were recently updated and will use PARPi monotherapy in patients with germline BRCA or PALB2 mutation after progression while receiving CDK4 / 6 inhibitors [69].
In the neoadjuvant setting, the percentage of pCR is very impressive with PARPi (52%), although it seems to be limited to the group with germline BRCA1/2.
Finally, the adjuvant benefit remains limited due to the low number of cases in the studies.
One of the most interesting data is that patients with BRCA1 / 2-mutated ER + / HER2- mBC have a higher risk of early disease progression while taking the CDK4 / 6 inhibitor. I think more should be specified now. which I consider a very crucial data
A section of conclusions would be missing where these data will be summarized: The percentage of almost 20% of potential patients candidates for PARPi, the benefit found in metastatic breast cancer and its limitation when they receive previous treatment with chemotherapy, especially the lower benefit found in this group of patients with cyclin inhibitors, and finally the role of PARPi in neoadjuvant and adjuvant
Reviewer 2 Report
This a really comprehensive review on current learnings on the favorable role of PARPi in patients with HER2-negative breast cancers disease and the importance of selecting the right patients that probably benefit from PARPi therapy. Authors illustrate the utility of PARPi in patients with BRCA alterations, but also highlighting the potential use in patients harboring additional alterations in HDR pathway. So, based on that, the evaluation of genes alterations in the patients is absolutely critical for maximizing PARPi as therapy.
Overall, the work is very well presented, easier to follow, and well connected with the clinical trials outcomes. I fully support the publication.
Reviewer 3 Report
The review by Collet et al discusses the recent introduction of PARP inhibitors in the treatment of advanced breast cancer, with a particular focus on hormone receptor positive HER2 negative breast cancer.
The review is written in correct English. The topic is of interest, reporting recently published literature data.
In my opinion, the review requires only very few minor revisions:
-in the "introduction", please explain more clearly the concept of "sinthetic lethality", in particular the accumulation of doulble-strand breaks which cannot be repaired by defective BRCA1 or BRCA2 proteins;
-in the description of the RUBY trial, I suggest to underline that enrolled patients were gBRCA WT mBC tested for LOH;
-in the description of the TBCRC048 trial, I suggest to explain in the text what is showed in figure 1.
